# EMBODIEDSAM: ONLINE SEGMENT ANY 3D THING IN REAL TIME

**Xiuwei Xu[1], Huangxing Chen[1], Linqing Zhao[1], Ziwei Wang[2], Jie Zhou[1], Jiwen Lu[1]***
[1]Tsinghua University, [2]Nanyang Technological University

## ABSTRACT

Embodied tasks require the agent to fully understand 3D scenes simultaneously with its exploration, so an *online*, *real-time*, *fine-grained* and *highly-generalized* 3D perception model is desperately needed. Since high-quality 3D data is limited, directly training such a model in 3D is infeasible. Meanwhile, vision foundation models (VFM) has revolutionized the field of 2D computer vision with superior performance, which makes the use of VFM to assist embodied 3D perception a promising direction. However, most existing VFM-assisted 3D perception methods are either offline or too slow that cannot be applied in practical embodied tasks. In this paper, we aim to leverage Segment Anything Model (SAM) for real-time 3D instance segmentation in an online setting. This is a challenging problem since future frames are not available in the input streaming RGB-D video, and an instance may be observed in several frames so efficient object matching between frames is required. To address these challenges, we first propose a geometric-aware query lifting module to represent the 2D masks generated by SAM by 3D-aware queries, which is then iteratively refined by a dual-level query decoder. In this way, the 2D masks are transferred to fine-grained shapes on 3D point clouds. Benefit from the query representation for 3D masks, we can compute the similarity matrix between the 3D masks from different views by efficient matrix operation, which enables real-time inference. Experiments on ScanNet, ScanNet200, SceneNN and 3RScan show our method achieves state-of-the-art performance among online 3D perception models, even outperforming offline VFM-assisted 3D instance segmentation methods by a large margin. Our method also demonstrates great generalization ability in several zero-shot dataset transferring experiments and show great potential in data-efficient setting. Code is available[1].

## 1 INTRODUCTION

Embodied tasks, like robotic manipulation and navigation Mousavian et al. (2019); Chaplot et al. (2020); Zhang et al. (2023), require the agent to understand the 3D scene, reason about human instructions and make decisions with self-action. Among the pipeline, embodied visual perception is the foundation for various downstream tasks. In embodied scenarios, we hope the 3D perception model to be: (1) *online*. The input data is a streaming RGB-D video rather than a pre-collected one and visual perception should be performed synchronously with data collection; (2) *real-time*. High inference speed is needed for robot planning and control; (3) *fine-grained*. It should recognize almost any object appeared in the scene; (4) *highly-generalized*. One model can be applied to different kinds of scenes and be compatible with different sensor parameters like camera intrinsics. As high-quality 3D data is limited, training such a model in pure 3D is almost infeasible.

Inspired by the great achievements of large language models (LLMs) Zhang et al. (2022); Chowdhery et al. (2023); Achiam et al. (2023), a series of vision foundation models (VFMs) such as SAM Kirillov et al. (2023) and SEEM Zou et al. (2023), have emerged. VFMs are revolutionizing the field of 2D computer vision by their fine-grained, accurate and generalizable segmentation on image pixels. However, less studies have been conducted on developing VFMs for the 3D domain. Since there is much less high-quality annotated 3D data compared with 2D counterparts, it holds great promise to

---

*Corresponding author.

[1]Project page: https://xuxw98.github.io/ESAM/

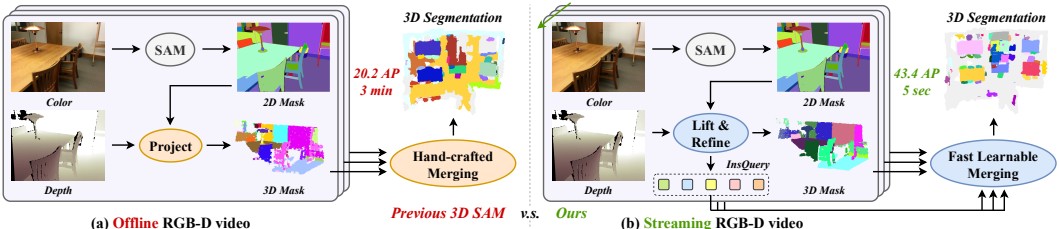

Figure 1: Different from previous 3D SAM methods Yang et al. (2023); Xu et al. (2023); Yin et al. (2024) that project 2D masks to 3D and merge them with hand-crafted strategies, ESAM lifts 2D masks to 3D queries and iteratively refine them to predict accurate 3D masks. With 3D queries, ESAM is also able to fastly merge 3D masks in different frames with simple matrix operations. Take SAM3D Yang et al. (2023) for comparison, our ESAM surpasses its performance by 23.2% AP with a more than $20\times$ faster speed.

explore the adaptation or extension of existing 2D VFMs for embodied 3D perception. Recently, there are some works Yang et al. (2023); Yin et al. (2024); Lu et al. (2023) that adopt SAM to automatically generate masks on multi-view images of a 3D scene and merge the masks in 3D with projection and iterative merging. While these approaches achieve fine-grained 3D instance segmentation with high generalization ability, they still face some serious problems that hinder their application: (1) they apply SAM on individual images and directly project the 2D masks to 3D point clouds with camera parameters. So the predictions are not geometric-aware, which may produce inconsistent results across different views; (2) they merge per-frame mask predictions in 3D with hand-crafted strategy. E.g., computing geometric similarity between all pairs of masks and merge them according to a threshold, which is inaccurate and very slow; (3) most of them are offline methods based on pre-collected RGB-D frames with 3D reconstruction, requiring full data collection before perception.

In this paper, we propose a VFM-assisted 3D instance segmentation framework namely Embodied-SAM (ESAM), which exploits the power of SAM to online segment anything in 3D scenes with high accuracy, fast speed and strong generalization ability. As shown in Figure 1, different from previous 3D SAM methods Yang et al. (2023); Xu et al. (2023); Yin et al. (2024) that project 2D masks to 3D and merge them with hand-crafted strategies, ESAM lifts 2D masks to 3D queries and predicts temporal and geometric-consistent 3D masks with iterative query refinement. Benefit from the 3D query representation, ESAM is also able to fastly merge 3D masks in different frames with simple matrix operations. Specifically, we extract point-wise features from the point clouds projected from depth image. Then we regard the 2D masks generated by SAM as superpoints, which is used to guide mask-wise aggregation by our proposed geometric-aware pooling module, generating 3D queries with one-to-one correspondence to SAM masks. We further present a dual-level query decoder to iteratively refine the 3D queries, which makes the queries efficiently attend with superpoint-wise features and generate fine-grained point-wise masks. Since each 3D instance mask is associated with a query, we can compute similarity between newly predicted 3D masks and previous ones by efficient matrix multiplication in parallel and accurately merge them. To enhance the discriminative ability of query features, we design three representative auxiliary tasks for estimation of geometric, contrastive and semantic similarities. We conduct extensive experiments on ScanNet, ScanNet200, SceneNN and 3RScan datasets. ESAM achieves both leading accuracy and speed among online 3D perception models. Compared with offline VFM-assisted 3D instance segmentation methods, we improve the performance by a large margin while still remain strong generalization ability. Moreover, ESAM also shows great potential in data-efficient setting when trained with limited data.

## 2 RELATED WORK

**VFM-assisted 3D Scene Segmentation:** In 2D realm, vision foundation models (VFM) Oquab et al. (2023); Kirillov et al. (2023); Li et al. (2023) have exploded in growth. Benefit from the large amount of annotated visual data, the 2D VFM shows great accuracy and very strong generalization ability, which makes them work well in zero-shot scenarios. Since there is much less high-quality annotated data in the field of 3D vision than the 2D counterpart, using 2D VFM to assist 3D scene perception

becomes a promising direction Rozenberszki et al. (2024); Yang et al. (2023); Yin et al. (2024). UnScene3D Rozenberszki et al. (2024) considers 2D self-supervised features from DINO Oquab et al. (2023) to generate initial pseudo masks, which is then iteratively refined with self-training. SAM3D Yang et al. (2023) adopts SAM Kirillov et al. (2023) to generate 2D instance masks, which are then projected to 3D space by depth and camera parameters and merged according to the geometry. SAI3D Yin et al. (2024) generates 3D primitives on the reconstructed 3D mesh. Then it adopts semantic-SAM to acquire 2D masks with semantic scores, which are connected with the 3D primitives and merged by a graph-based region growing strategy. Recently, there are some works Qin et al. (2024); Ye et al. (2024) leverage VFM to segment 3D scenes represented by 3D gaussian, which provide a new perspective for 3D instance segmentation. Our approach also utilizes SAM to assist 3D instance segmentation. Differently, we makes the process of 2D-to-3D projection and 3D mask merging learnable and online. In this way, our ESAM is able to predict more accurate 3D masks and be applied in practical real-time online tasks.

**Online 3D Scene Perception:** In persuit of embodied AI, real world applications like robotic navigation Chaplot et al. (2020); Zhang et al. (2023) and manipulation Mousavian et al. (2019) have received increasing attention. Online 3D scene perception, which precisely understands the surrounding 3D scenes from streaming RGB-D videos, becomes the visual basis of these robotic tasks. Early online 3D perception methods process 2D images separately and project the predictions to 3D point clouds, which is followed by a fusion step to merge the predictions from different frames McCormac et al. (2017); Narita et al. (2019). However, the predictions on 2D image is not geometric and temporal-aware, which makes the fusion step difficult and inaccurate. Fusion-aware 3D-Conv Zhang et al. (2020) and SVCNN Huang et al. (2021) construct data structures to maintain the information of previous frames and conduct point-based 3D aggregation to fuse the 3D features for semantic segmentation. INS-CONV Liu et al. (2022) extends sparse convolution Graham et al. (2018); Choy et al. (2019) to incremental CNN to efficiently extract global 3D features for semantic and instance segmentation. In order to simplify the design of online 3D perception model and leverage the power of the advanced offline 3D architectures, MemAda Xu et al. (2024) proposes a new paradigm for online 3D scene perception, which empowers offline model with online perception ability by multimodal memory-based adapters. Different from the previous works, our ESAM lifts SAM-generated 2D masks to accurate 3D mask and corresponding queries, which enables us to efficiently merge the per-frame predictions with high accuracy.

## 3 APPROACH

Given a sequence of RGB-D images $\mathcal{X}_t = \{x_1, x_2, ..., x_t\}$ with known poses, our goal is to segment any instance in the corresponding 3D scene. Formally, $x_t = (I_t, P_t)$ where $I_t$ is the color image and $P_t$ is the point clouds acquired by projecting the depth image to 3D space with pose parameters. Our method is required to predict instance masks for the observed 3D scene $S_t = \bigcup_{i=1}^{t} P_i$. Furthermore, we want to solve this problem online; that is, at any time instant $t$ future frames $x_i$, $i > t$ are not known, and temporally consistent 3D instance masks of $S_t$ should be predicted at each time instant.

**Overview.** The overview of our approach is shown in Figure 2. We solve the problem of online 3D instance segmentation in an incremental manner to achieve real-time processing. At time instant $t$, we only predict the instance masks $M_t^{cur}$ of current frame $P_t$. Then we merge $M_t^{cur}$ to the previous instance masks $M_{t-1}^{pre}$ of $S_{t-1}$ and get the updated instance masks $M_t^{pre}$ of $S_t$.

### 3.1 QUERY LIFTING AND REFINEMENT

Consider the model is receiving the $t$-th RGB-D frame $x_t = (I_t, P_t)$, we first adopt SAM automatic mask generation to acquire 2D instance masks $M_t^{2d}$ from $I_t$. In this subsection, we ignore the subscript $t$ for clearer statement.

**Geometric-aware Query Lifting.** As SAM does not leverage the information from previous frames, nor does it exploit 3D information from the depth image, directly project $M^{2d}$ to $P$ results in inaccurate and temporal-inconsistent 3D masks. Instead, we aim to lift each 2D mask to a 3D query feature, which enables us to further refine the queries for 3D instance mask generation. Since the 2D binary mask is less informative, here we propose to extract point cloud features from the scene and then regard the 2D masks as indexs to cluster point clouds into superpoints, where queries can

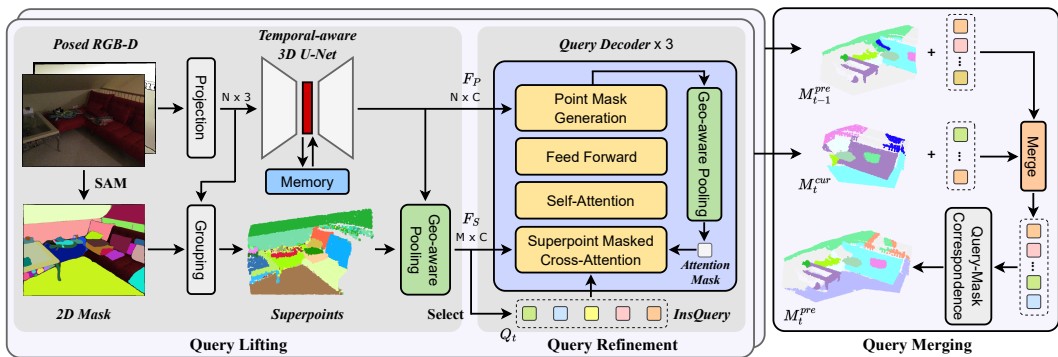

Figure 2: Overview of *ESAM*. At a new time instant $t$, we first adopt SAM to generate 2D instance masks $M_t^{2d}$. We propose a geometric-aware query lifting module to lift $M_t^{2d}$ to 3D queries $Q_t$ while preserving fine-grained shape information. $Q_t$ are refined by a dual-level decoder, which enables efficient cross-attention and generates fine-grained point-wise masks $M_t^{cur}$ from $Q_t$. Then $M_t^{cur}$ is merged into previous masks $M_{t-1}^{pre}$ by a fast query merging strategy.

be simply selected from the superpoint features. Assuming the point clouds $P \in \mathbb{R}^{N \times 3}$ and there are $M$ masks in $M^{2d}$, we first map $M^{2d}$ to $P$ according to the color-depth correspondence to get superpoint index $S \in \mathbb{Z}^N$, where each element in $S$ falls in $[0, M)$. Then we feed $P$ to a 3D sparse U-Net Choy et al. (2019) with memory-based adapter Xu et al. (2024) to extract temporal-aware 3D features $F_P \in \mathbb{R}^{N \times C}$. With $F_P$ and $S$, we can pool the point-wise features to superpoint features $F_S \in \mathbb{R}^{M \times C}$.

However, naive operation such as max or average pooling may degrade the representation ability of $F_S$. To better preserve the point features inside each superpoint, we take the geometric shape of each superpoint into account. For a superpoint $P^i \subseteq P$, $i \in [0, M)$, we compute the normalized relative positions $p_j^r$ of all points $p_j \in P^i$ with respect to the superpoint's center $c_i$. In this way, the set $\mathcal{P}_i = \{p_j^r = \frac{p_j - c_i}{\max(p_j) - \min(p_j)} \mid p_j \in P^i\}$ represents the normalized shape of this superpoint with diameter of 1 and center of origin. Then we compute the local and global features for each point:

$$\boldsymbol{z}^{global} = \mathrm{Agg}(\boldsymbol{z}^{local}) \in \mathbb{R}^C, \; \boldsymbol{z}^{local} = \mathrm{MLP}(\mathcal{P}_i) \in \mathbb{R}^{|\mathcal{P}_i| \times C} \quad (1)$$

where MLP performs on each individual point and Agg is the aggregation function implemented with channel-wise max-pooling. The local and global features represent the relevance between points and shape, so we concat both features and feed them to another MLP to predict point-wise weight:

$$w_j = \mathrm{Sigmoid}(\mathrm{MLP}(\boldsymbol{z}_j)) \in \mathbb{R}^{(0,1)}, \; \boldsymbol{z}_j = [\boldsymbol{z}_j^{local}, \boldsymbol{z}^{global}] \quad (2)$$

Finally, we aggregate point features $F_P^i$ into the $i$-th superpoint with weighted average pooling:

$$F_S^i = \mathcal{G}(F_P^i) + \boldsymbol{z}^{global}, \; \mathcal{G}(F_P^i) = \mathrm{mean}(F_P^i * [w_1, ..., w_{|\mathcal{P}_i|}]) \quad (3)$$

Note we enhance the pooled superpoint feature with $\boldsymbol{z}^{global}$ to fully combine the shape-level geometric feature and scene-level 3D U-Net feature. The computation for each superpoint can be parallelized with point-wise MLP and Scatter function Fey, so this geometric-aware pooling is actually efficient.

**Dual-level Query Decoder.** After pooling, the $M$ 2D instance masks $M^{2d}$ are lifted to 3D superpoint features $F_S$. Then we initialize a series of 3D instance queries $Q_0$ from $F_S$, which are iteratively refined by several transformer-based query decoder layers and leveraged to predict 3D masks. During training, we randomly sample a proportion between 0.5 and 1 of $F_S$ to construct $Q_0$ for data augmentation. While at inference time we simply set $Q_0 = F_S$.

Each qeury decoder employs masked cross-attention between queries and the scene representations to aggregate instance information for each query:

$$\hat{Q}_l = \mathrm{Softmax}(\frac{\boldsymbol{Q} \cdot \boldsymbol{K}^T}{\sqrt{C}} + A_l) \cdot \boldsymbol{V}, \; A_l(i,j) = \begin{cases} 0 & \text{if } M_l^{cur}(i,j) = \text{True} \\ -\infty & \text{otherwise} \end{cases}, \; l = 0,1,2 \quad (4)$$

where $\cdot$ indicates matrix multiplication, $\boldsymbol{Q}$ is the linear projection of $Q_l$, $\boldsymbol{K}$ and $\boldsymbol{V}$ are the linear projection of the scene representations $F$. $F$ can be point-wise features $F_P$ or superpoint-wise features $F_S$. $A_l$ is the attention mask derived from the predicted 3D instance masks $M_l^{cur}$ in the $l$-th decoder layer. $(i, j)$ indicates $i$-th query attending to $j$-th point or superpoint. Then we feed $\hat{Q}_l$ to self-attention layer and feed forward network to get $Q_{l+1}$, followed by a mask generation module to predict the instance mask for each query:

$$M_l^{cur} = \text{Sigmoid}(\phi(Q_l) \cdot F^T) > \varphi, \ l = 0, 1, 2, 3 \tag{5}$$

where $\phi$ is a linear layer. $M_l^{cur}$ is a point mask if $F = F_P$, otherwise it is a superpoint mask.

A common practice Schult et al. (2022); Sun et al. (2023); Kolodiazhnyi et al. (2024b) for qeury decoder is to adopt the same level of scene representations for cross-attention and mask generation. However, since SAM has already outputs high-level semantic-aware masks, we observe $M \ll N$. If we adopt the point-wise scene representations $F_P$ for query decoder, the cross-attention operation will be memory-consuming due to the large amount of points. While if we use superpoint features $F_S$, the predicted 3D instance masks will only be the combination of superpoints and thus cannot be refined to finer granularity. To get the best of both worlds, our query decoder is designed to be dual-level. For cross-attention in Eq (4), we set $F = F_S$ to achieve efficient interaction. While for mask prediction in Eq (5), we set $F = F_P$ for fine-grained mask generation. To support masked attention, we pool point mask to superpoint mask before Eq (4):

$$M_l^{cur} \leftarrow \mathcal{G}(M_l^{cur}) > \varphi \tag{6}$$

where $\mathcal{G}$ is the geometric-aware pooling in Eq (3). We can reuse the pre-computed weights in Eq (2) to reduce computation. In this way, after $3\times$ query decoders, we acquire accurate point masks $M_3^{cur}$ as well as the corresponding queries $Q_3$, which is denoted as $M_t^{cur}$ and $Q_t$ in the following subsections. We perform mask-NMS on $M_t^{cur}$ to filter out redundant masks as well as the corresponding queries.

## 3.2 Efficient Online Query Merging

Once lifting 2D masks $M_t^{2d}$ to accurate 3D masks $M_t^{cur}$, we then merge $M_t^{cur}$ to the previous instance masks $M_{t-1}^{pre}$ to acquire $M_t^{pre}$. Note when $t = 1$ we have $M_1^{pre} = M_1^{cur}$ as an initialization.

However, the mainstream solution for merging instance masks in previous works Yang et al. (2023); Yin et al. (2024); Liu et al. (2022); Narita et al. (2019); Lu et al. (2023) is to traverse over all masks in $M_t^{cur}$ and compare each mask in $M_t^{cur}$ with all previous masks in $M_{t-1}^{pre}$. This process is very slow, because in order to accurately decide wheter a new mask shoule be merged into a previous mask, the geometric similarity such as mask-IoU or CD-distance is computed on the point clouds of the two masks. The computation of similarity involves all the points in each mask, which has high computation complexity. What is worse, above operations are hard to be computed in parallel, since the number of points in each mask is different and we need to pick out point clouds of each instance according to the mask one by one. To this end, we propose to represent each mask in fixed-size vectors and compute the similarity with efficient matrix operation.

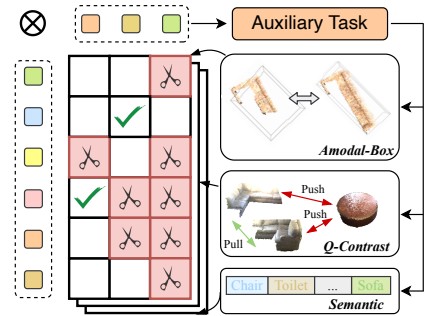

Figure 3: Details of our efficient query merging strategy. We propose three kinds of representative auxiliary tasks, which generates geometric, contrastive and semantic representations in the form of vectors. Then the similarity matrix can be efficiently computed by matrix multiplication. We further prune the similarity matrix and adopt bipartite matching to merge the instances.

Benefit from our architecture, for each mask in $M_t^{cur}$ and $M_{t-1}^{pre}$ we have the corresponding query feature. The query feature itself is a fixed-size vector representation, but simply computing similarity between them is less informative, which gets very low performance. Therefore, we set up several representative auxiliary tasks based on the query features to learn vector representations under different metrics, which are used to compute *geometric*, *contrastive* and *semantic* similarities.

First, for geometric similarity, we observe the model is able to learn the whole geometry with only partial observation. However, due to the restriction of segmentation that predictions can only be made

on existing points, the model cannot express its knowledge about the whole geometry. Therefore, we make the model able to express its full knowledge by introducing an auxiliary task of bounding box prediction. We adopt a MLP to predict the bounding box regression based on the center of each query (i.e. the center $c_i$ of the corresponding superpoint) to get box $B \in \mathbb{R}^6$. Then the geometric similarity between two masks can be computed by the IoU between the two boxes. We ignore box orientations since the IoU matrix between two sets of axis-aligned bounding boxes can be computed by simple matrix operation.

Second, for contrastive similarity, we aim to learn an instance-specific representation where features from the same instance should be pulled together and otherwise pushed away. This representation can be learned by contrastive training between two adjacent frames: we use MLP to map the query features $Q_t$ to contrastive feature $f_t$. Then for an instance $i$ appears in the $t$-th and $(t+1)$-th frames, we choose the two features of this instance $(f_t^i, f_{t+1}^i)$ as the positive pair, and sample features from other instances $(f_t^i, f_{t+1}^k)$ as the negative pair. The detailed loss function is shown in the next subsection.

Finally, for semantic similarity, we simply adopt a MLP to predict per-category probability distribution $S \in \mathbb{R}^K$, where $K$ is the number of pre-defined categories. There are also other choices for this task. For example, if we adopt semantic-SAM Li et al. (2023) instead of SAM, we can directly utilize the semantic predictions for the 2D masks to serve as $S$ for the corresponding queries.

In this way, the similarity matrix $\mathcal{C}$ between $M_{t-1}^{pre}$ and $M_t^{cur}$ can be efficiently computed with their corresponding geometric, contrastive and semantic representations:

$$\mathcal{C} = \text{IoU}(B_{t-1}^{pre}, B_t^{cur}) + \frac{f_{t-1}^{pre}}{||f_{t-1}^{pre}||_2} \cdot \left(\frac{f_t^{cur}}{||f_t^{cur}||_2}\right)^T + \frac{S_{t-1}^{pre}}{||S_{t-1}^{pre}||_2} \cdot \left(\frac{S_t^{cur}}{||S_t^{cur}||_2}\right)^T \tag{7}$$

where $\text{IoU}(\cdot, \cdot)$ means the IoU matrix between two set of axis-aligned bounding boxes. We prune $\mathcal{C}$ by setting elements smaller than threshold $\epsilon$ to $-\infty$. Then bipartite matching with cost $-\mathcal{C}$ is performed on $M_{t-1}^{pre}$ and $M_t^{cur}$, which assigns each mask in $M_t^{cur}$ to one of the masks in $M_{t-1}^{pre}$. If a new mask fails to match with any previous mask, we register a new instance for this mask. Otherwise we merge the two masks as well as their $B$, $f$ and $S$. Mask merging can be simply implemented by taking union. While for other representations, we weighted average them by: $B_t^{pre}[i] = \frac{n}{n+1}B_{t-1}^{pre}[i] + \frac{1}{n+1}B_t^{cur}[j]$ and so on. We assume the $j$-th new mask is merged to the $i$-th previous mask. $n$ is the count of merging, which indicates the number of masks that have been merged to $M_{t-1}^{pre}[i]$.

## 3.3 LOSS FUNCTION

We have semantic and instance labels on each RGB-D frame. In each RGB-D video, the instance labels of different frames are consistent. Given the annotations, we compute per-frame losses based the predictions from each query. Since the queries $Q_t$ are lifted from 2D SAM masks in a one-to-one way, we ignore the complicated label assignment step and directly utilize the annotations on 2D mask to supervise the predictions from the corresponding query. We assume that a 2D SAM mask can belong only to one instance, and thus we can acquire the ground-truth semantic label and 2D instance mask for each query. We utilize the pixel correspondence with depth image to map 2D instance mask to 3D point clouds, and compute ground-truth axis-aligned bounding box based on the 3D instance mask. With above annotations, we compute binary classification loss $\mathcal{L}_{cls}^t$ with cross-entropy to discriminate foreground and background instances. The predicted 3D mask is supervised by a binary cross-entropy $\mathcal{L}_{bce}^t$ and a Dice loss $\mathcal{L}_{dice}^t$. The losses for bounding boxes and semantic predictions are defined as IoU-loss $\mathcal{L}_{iou}^t$ and binary cross-entropy $\mathcal{L}_{sem}^t$ respectively.

Apart from the above per-frame losses, we also formulate a contrastive loss between adjacent frames:

$$\mathcal{L}_{cont}^{t \to t+1} = -\frac{1}{Z}\sum_{i=1}^{Z} \log \frac{e^{(\langle f_t^i, f_{t+1}^i \rangle / \tau)}}{\sum_{j \neq i} e^{(\langle f_t^i, f_{t+1}^j \rangle / \tau)} + e^{(\langle f_t^i, f_{t+1}^i \rangle / \tau)}} \tag{8}$$

where $\langle \cdot, \cdot \rangle$ is cosine similarity. So finally the total loss is formulated as:

$$\mathcal{L} = \frac{1}{T}\sum_{t=1}^{T}(\alpha\mathcal{L}_{cls}^t + \mathcal{L}_{bce}^t + \mathcal{L}_{dice}^t + \beta\mathcal{L}_{iou}^t + \mathcal{L}_{sem}^t + \mathcal{L}_{cont}^{t \to t+1} + \mathcal{L}_{cont}^{t \to t-1}) \tag{9}$$

where $\mathcal{L}_{cont}^{T \to T+1}$ and $\mathcal{L}_{cont}^{1 \to 0}$ is set to 0.

Table 1: Class-agnostic 3D instance segmentation results of different methods on ScanNet200 dataset. The unit of Speed is $ms$ per frame, where the speed of VFM and other parts are reported separately.

| Method | Type | VFM | AP | $AP_{50}$ | $AP_{25}$ | Speed |
|--------|------|-----|-----|-----------|-----------|-------|
| SAMPro3D | Offline | SAM | 18.0 | 32.8 | 56.1 | – |
| Open3DIS | Offline | GroundedSAM | 34.6 | 43.1 | 48.5 | – |
| SAI3D | Offline | SemanticSAM | 28.2 | 47.2 | 67.9 | – |
| SAM3D | Online | SAM | 20.2 | 35.7 | 55.5 | 1369+1518 |
| ESAM | Online | SAM | 42.2 | 63.7 | 79.6 | 1369+**80** |
| ESAM-E | Online | FastSAM | **43.4** | **65.4** | **80.9** | 20+**80** |

Table 2: Dataset transfer results of different methods from ScanNet200 to SceneNN and 3RScan. We directly evaluate the models in Table 1 on other datasets to show their generalization ability.

| Method | Type | ScanNet200→SceneNN | | | ScanNet200→3RScan | | |
|--------|------|-----|-----------|-----------|-----|-----------|-----------|
| | | AP | $AP_{50}$ | $AP_{25}$ | AP | $AP_{50}$ | $AP_{25}$ |
| SAMPro3D | Offline | 12.6 | 25.8 | 53.2 | 3.9 | 8.0 | 21.0 |
| Open3DIS | Offline | 18.2 | 32.2 | 48.9 | 9.5 | 21.8 | 47.0 |
| SAI3D | Offline | 18.6 | 34.7 | 65.7 | 8.1 | 16.9 | 37.0 |
| SAM3D | Online | 15.1 | 30.0 | 51.8 | 6.2 | 13.0 | 33.9 |
| ESAM | Online | **28.8** | **52.2** | 69.3 | **14.1** | **31.2** | **59.6** |
| ESAM-E | Online | 28.6 | 50.4 | **71.0** | 13.9 | 29.4 | 58.8 |

## 4 EXPERIMENT

In this section, we first describe our datasets and implementation details. Then we compare our method with state-of-the-art VFM-assisted 3D instance segmentation methods online 3D segmentation methods to validate its effectiveness. We also apply ESAM in data-efficient setting to demonstrate its application potential. Finally we conduct ablation studies to provide a comprehensive analysis on our design. More supplementary experiments can be found in appendix.

### 4.1 BENCHMARK AND IMPLEMENTATION DETAILS

We evaluate our method on four datasets: ScanNet Dai et al. (2017), ScanNet200 Rozenberszki et al. (2022), SceneNN Hua et al. (2016) and 3RScan Wald et al. (2019). ScanNet contains 1513 scanned scenes, out of which we use 1201 sequences for training and the rest 312 for testing. ScanNet200 provides more fine-grained annotations on the scenes of ScanNet, which contains more than 200 categories. SceneNN contains 50 high-quality scanned scenes with instance and semantic labels. Following Xu et al. (2024), we select 12 clean sequences for testing. 3RScan is a more challenging indoor dataset where the RGB-D sequences are acquired by fast-moving cameras. We choose its test split for testing, which contains 46 scenes. Each dataset provide both posed RGB-D sequences and reconstructed point clouds with labels.

**Benchmarks:** We compare different methods on four benchmarks. First, we compare with VFM-assisted 3D instance segmentation methods in Table 1. We train different methods on ScanNet200 training set (if needed) and evaluate them on ScanNet200 validation set in a class-agnostic manner. We also train and evaluate the methods in the same way on ScanNet, where the results are put in appendix (Table 7). For offline methods, the input of each scene is a reconstructed point cloud and a RGB-D video, where predictions are made on the reconstructed point clouds. For online methods, the input is a streaming RGB-D video, and we map the final predicted results on $S_t$ to the reconstructed point clouds with nearest neighbor interpolation for comparison.

Since zero-shot methods like SAM3D do not require training. To fairly compare them with learnable methods, we further evaluate the models in Table 1 on SceneNN and 3RScan without finetuning. This benchmark, shown in Table 2, validates the generalization ability of different methods.

Table 3: 3D instance segmentation results of different methods on ScanNet and SceneNN datasets.

| Method | Type | ScanNet | | | SceneNN | | | FPS |
| --- | --- | --- | --- | --- | --- | --- | --- | --- |
| | | AP | $AP_{50}$ | $AP_{25}$ | AP | $AP_{50}$ | $AP_{25}$ | |
| TD3D | Offline | 46.2 | 71.1 | 81.3 | – | – | – | – |
| Oneformer3D | Offline | 59.3 | 78.8 | 86.7 | – | – | – | – |
| INS-Conv | Online | – | 57.4 | – | – | – | – | – |
| TD3D-MA | Online | 39.0 | 60.5 | 71.3 | 26.0 | 42.8 | 59.2 | 3.5 |
| ESAM-E | Online | 41.6 | 60.1 | 75.6 | 27.5 | 48.7 | **64.6** | **10.0** |
| ESAM-E+FF | Online | **42.6** | **61.9** | **77.1** | **33.3** | **53.6** | 62.5 | 9.8 |

Finally, we compare with online 3D instance segmentation methods in Table 3. Following previous works Liu et al. (2022); Xu et al. (2024), we train different methods on ScanNet training set and evaluate them on ScanNet validate set and SceneNN.

**Compared methods:** On Table 1 and Table 2, we compare ESAM with SAM3D Yang et al. (2023), SAMPro3D Xu et al. (2023), Open3DIS Nguyen et al. (2024) and SAI3D Yin et al. (2024). We adopt the 2D version of Open3DIS for a fair comparison. Because its 3D version makes predictions directly on reconstructed point clouds, while other methods only make predictions on RGB-D frames. On Table 3, we compare with TD3D Kolodiazhnyi et al. (2024a), Oneformer3D Kolodiazhnyi et al. (2024b), INS-Conv Liu et al. (2022) and TD3D-MA Xu et al. (2024). In terms of VFM, above methods mainly adopt SAM Kirillov et al. (2023), GroundedSAM Ren et al. (2024), SemanticSAM Li et al. (2023) and FastSAM Zhao et al. (2023).

**Implementation details:** Following Xu et al. (2024), we train ESAM in two stages. First we train a single-view perception model on ScanNet(200)-25k, which is a subset of ScanNet(200) with individual RGB-D frames, without memory-based adapters and losses for the three auxiliary tasks. Next we finetune the single-view perception model on RGB-D sequences with the adapters and full losses. To reduce memory footprint, we randomly sample 8 adjacent RGB-D frames for each scene at every iteration. For hyperparameters, we set $\varphi = 0.5$, $\epsilon = 1.75$, $\tau = 0.02$, $\alpha = 0.5$ and $\beta = 0.5$. In the dual-level query decoder, we actually set $F = F_S$ for the first two iterations of mask prediction, and then set $F = F_P$. This smoothens the mask generation process with curriculum learning.

## 4.2 COMPARISON WITH STATE-OF-THE-ART

We compare our method with the top-performance VFM-assisted 3D instance segmentation methods and online 3D instance segmentation methods as described above. We provide three versions of ESAM, namely ESAM, ESAM-E and ESAM-E+FF. ESAM adopts SAM as the VFM while ESAM-E adopts FastSAM to achieve real-time inference. ESAM-E+FF not only adopts the 2D masks from FastSAM, but also fuses image features extracted by FastSAM backbone to point clouds following Rukhovich et al. (2023). We also include some visualization results in for qualitative evaluation.

According to Table 1, on class-agnostic 3D instance segmentation task (i.e. the 3D "segment anything task"), our ESAM establishes new state-of-the-art compared with previous methods, even including the offline ones. Note that it is much more challenging for online methods to perceive the 3D scenes compared to offline alternatives, since offline methods directly process the complete reconstructed 3D scenes while online methods deal with partial and noisy frames. Despite the high accuracy, ESAM is also much faster than previous methods. It takes only 80ms to process a frame due to the efficient architecture design and fast merging strategy, while methods like SAM3D that adopts hand-crafted merging strategy requires more than 1s per frame. When replacing SAM with the faster alternative FastSAM, ESAM-E can achieve real-time online 3D instance segmentation with about 10 FPS, while the accuracy is still much higher than previous methods.

In terms of generalization ability, ESAM also demonstrates great performance. As shown in Table 2, when directly transferred to other datasets, ESAM still achieves state-of-the-art accuracy compared with zero-shot methods. We note offline methods perform worse on 3RScan dataset, this is because they highly rely on clean reconstructed 3D meshes with accurately aligned RGB frames. While in 3RScan, the camera is moving fast and thus the RGB images and camera poses are blurry.

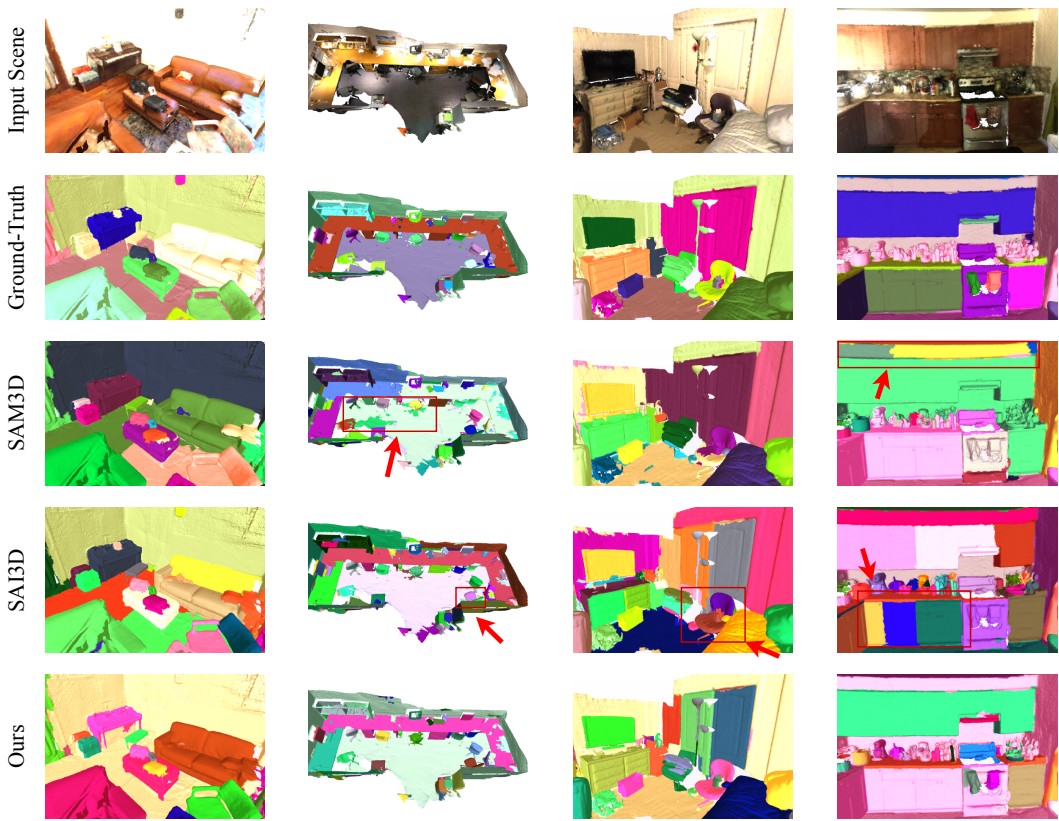

Figure 4: Visualization results of different 3D instance segmentation methods on ScanNet200 dataset. As highlighted in red boxes, SAM3D predicts noisy masks while SAI3D tends to over segment an instance into multiple parts.

We visualize the predictions of the above methods on ScanNet200, as shown in Figure 4. ESAM can predict accurate and fine-grained 3D instance segmentation masks, while being able to process streaming RGB-D video in real time. We also provide an online visualization to further demonstrate the practicability of ESAM in Figure 6 in appendix. More details can be viewed in our video demo.

As shown in Table 3, ESAM also achieves state-of-the-art performance compared with previous online 3D instance segmentation methods. Different from previous methods that only fuse 2D features to 3D point clouds, our approach utilize both 2D features and 2D masks to better guide the learning of 3D representation.

## 4.3 ANALYSIS OF ESAM

**Data-efficient learning.** We reduce the training samples by using only 20% or 50% training set and report the class-agnostic performance of ESAM on Scan-Net200 in Table 4. It is shown that the performance degradation of ESAM is not significant with only half the training data. Moreover, ESAM still achieves state-of-the-art performance even with 10% training data (SAI3D: 28.2/47.2/67.9). This is because 2D VFM has already provided a good initialization, thus the learning part of ESAM is easy to converge.

Table 4: Performance of ESAM when trained with partial training set.

| Proportion | AP | $AP_{50}$ | $AP_{25}$ |
|---|---|---|---|
| 100% | 42.2 | 63.7 | 79.6 |
| 50% | 40.2 | 62.3 | 78.4 |
| 10% | 32.8 | 54.1 | 73.9 |

**Ablation study.** We conduct ablation studies to validate the effectiveness of the proposed methods. For architecture design, we conduct experiments on ScanNet-25k and report AP and average inference latency (ms) of each frame excluding VFM in Table 5. It can be seen that geometric-aware pooling boosts the performance up to 1.3% while brings negligible computational overhead. Note that the prediction error on single views will accumulate on the whole scenes, so a high AP on ScanNet-25k

contributes a lot to the final performance. We can also observe that the dual-level design in ESAM achieves comparable accuracy compared with the time-consuming $F = F_P$ strategy, while only slightly increases the latency compared with the fully-superpoint $F = F_S$ strategy. For the merging strategies, we compare different design on ScanNet with AP reported, as shown in Table 6. It is shown that each auxiliary task is important for the quality of mask merging. We notice that the geometric similarity has the most significant influence on the final performance. This is because most mask pairs can be excluded based on distance.

**Visualization of auxiliary tasks.** We also visualize the predictions of our auxiliary tasks for comprehensive understanding of ESAM. From Figure 5 (a), it can be observed that the model is able to predict the whole geometry of objects with only partial observation. The t-SNE visualization in Figure 5 (b) validates that the model successfully learns discriminative query representation for object matching. Finally the semantic segmentation results in Figure 5 (c) shows that our ESAM can learn satisfactory semantic representation and is extendable to 3D semantic segmentation task.

Table 5: Effects of the architecture design.

| Method | AP | Latency |
| --- | --- | --- |
| Replace $\mathcal{G}$ with average pooling | 45.9 | 43.6 |
| Set $F = F_S$ only | 34.5 | 43.1 |
| Set $F = F_P$ only | 47.4 | 51.7 |
| **The final model** | 47.2 | 45.4 |

Table 6: Effects of the merging strategies.

| Method | AP |
| --- | --- |
| Remove box representation | 33.4 |
| Remove contrastive representation | 36.9 |
| Remove semantic representation | 37.6 |
| **The final model** | 41.6 |

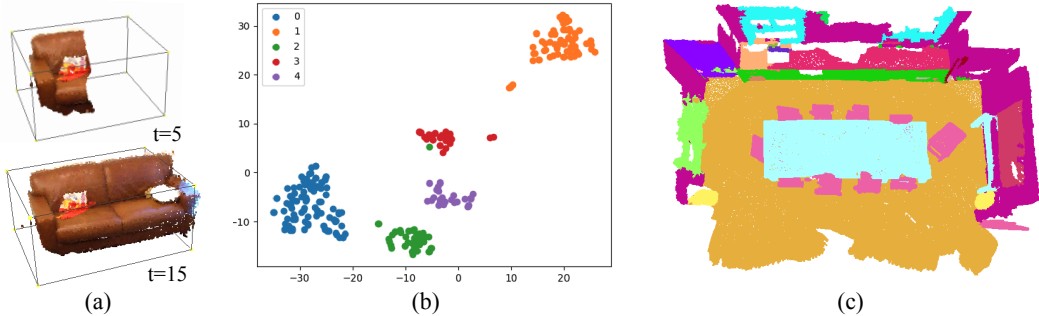

(a)  (b)  (c)

Figure 5: Visualization of the auxiliary tasks for our merging strategy. (a) 3D box prediction for geometric similarity. We visualize the bounding boxes of an object at different time instant. (b) t-SNE visualization of the instance-specific representation for contrastive similarity. Different colors indicate different instances and different points indicate the instance feature at different frames. (c) Query-wise semantic segmentation for semantic similarity.

## 5 CONCLUDING REMARK

In this work, we presented ESAM, an efficient framework that leverages vision foundation models for online, real-time, fine-grained and generalized 3D instance segmentation. We propose to lift the 2D masks generated by VFM to 3D queries with geometric-aware pooling, which is followed by a dual-path query decoder to refine the queries and generate accurate 3D instance masks. Then with the query-mask correspondence, we design three auxiliary tasks to discriminatively represent each 3D mask, which enables fast mask merging with matrix operations. Extensive experimental results on four datasets demonstrates that ESAM achieves leading performance, online and real-time inference and strong generalization ability. We believe ESAM brings a new paradigm on how to effectively leverage 2D VFM for embodied perception.

**Potential Limitations.** Despite of the satisfactory performance, there are still some limitations of ESAM. First, whether ESAM is real-time depends on the adopted VFM. Currently we adopt SAM and FastSAM, among which only FastSAM can achieve real-time inference. However, we believe there will be more efficient 2D VFM with better performance and more functions in the near future, and ESAM can be further improved along with the improvement of 2D VFM. Second, the 3D U-Net and memory-based adapters for feature extraction are relatively heavy, which count for most of the inference time for 3D part of ESAM. The speed of ESAM may be boosted to a higher level if we can make the backbone more efficient, which we leave for future work.

## ACKNOWLEDGEMENTS

This work was supported in part by the National Natural Science Foundation of China under Grant 624B2076, Grant 62125603, Grant 62336004, and Grant 62321005, and in part by the Beijing Natural Science Foundation under Grant No. L247009.

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

## A  Appendix

We provide more experimental results in the appendix.

### A.1  Online Visualization

We demonstrate the online 3D segmentation process in Figure 6. It is shown that ESAM can effectively merge partial segmentation results into a whole object and generate fine-grained 3D masks for the online reconstructed 3D scene.

### A.2  Additional Experiments

Following Yin et al. (2024), we compare ESAM with conventional clustering methods McInnes & Healy (2017); Nunes et al. (2022); Felzenszwalb & Huttenlocher (2004); Rozenberszki et al. (2024); Caron et al. (2021) and VFM-assisted 3D scene perception methods on ScanNet in Table 7. Consistent with the results in Table 1, ESAM also achieves leading performance and speed on ScanNet in the class-agnostic 3D instance segmentation setting.

### A.3  Analysis on Inference Time

We decompose the inference time of ESAM excluding VFM in Table 8. The temporal-aware backbone consists of a sparse convolutional U-Net and several memory-based adapters. The merging process consists of similarity computation, bipartite matching and mask / representation updating. Due to the efficient design, the decoder and merging operation of ESAM only take a small proportion of

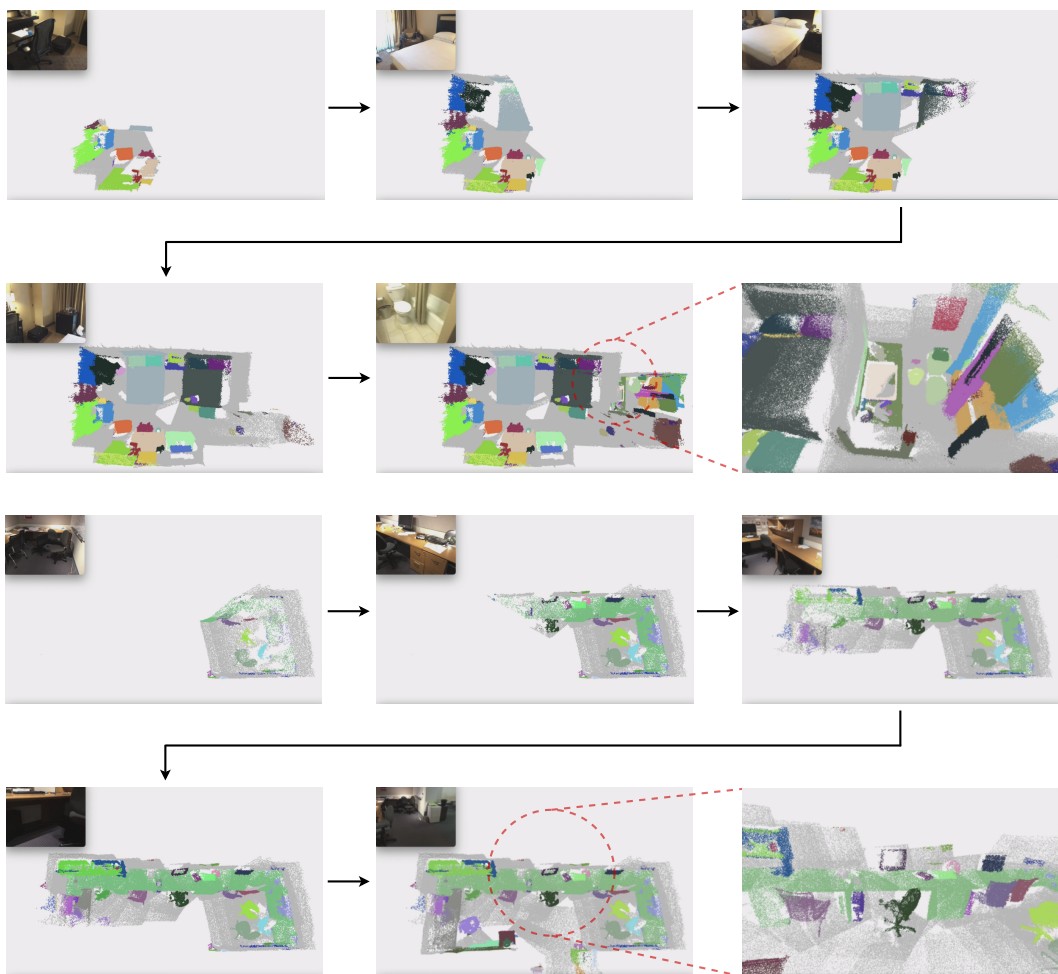

Figure 6: Online visualization of ESAM on ScanNet200 dataset. Refer to the video demo in our project page for more details.

inference time. In terms of backbone, it is promising to adopt techniques like network pruning or knowledge distillation Li et al. (2016); Hou et al. (2022) to accelerate its inference speed, which we leave for future work.

## A.4 OPEN-VOCABULARY RESULTS

We further evaluate the open-vocabulary 3D instance segmentation ability on the 198 categories of Scan-Net200 in Table 9. Since ESAM outputs class-agnostic 3D masks, there are two methods to extend it to open-vocabulary 3D segmentation. The first is to feed the class-agnostic 3D masks to open-vocabulary mask classification model like OpenMask3D Takmaz et al. (2023) and OpenIns3D Huang et al. (2024), which is adopted in the code of SAI3D Yin et al. (2024). The second is to adopt open-vocabulary 2D segmentation model to acquire the category labels for each 2D mask. Since there is one-to-one correspondence between 3D mask and 2D mask in ESAM, we can acquire the category labels for each 3D mask accordingly. Here we follow SAI3D to adopt the first method and compare with it.

Table 9: Open-vocabulary 3D instance segmentation results on ScanNet200 dataset.

| Method | AP | $AP_{50}$ | $AP_{25}$ |
|--------|------|------|------|
| SAI3D  | 9.6  | 14.7 | 19.0 |
| ESAM   | **13.7** | **19.2** | **23.9** |

Table 7: Class-agnostic 3D instance segmentation results of different methods on ScanNet dataset. Following Yin et al. (2024), we compare with conventional clustering methods and VFM-assisted 3D scene perception methods. The unit of Speed is $ms$ per frame, where the speed of VFM and other parts are reported separately.

| Method | Type | VFM | AP | $AP_{50}$ | $AP_{25}$ | Speed |
|---|---|---|---|---|---|---|
| HDBSCAN | Offline | – | 1.6 | 5.5 | 32.1 | – |
| Nunes et al. | Offline | – | 2.3 | 7.3 | 30.5 | – |
| Felzenszwalb et al. | Offline | – | 5.0 | 12.7 | 38.9 | – |
| UnScene3D | Offline | DINO | 15.9 | 32.2 | 58.5 | – |
| SAMPro3D | Offline | SAM | 16.7 | 31.5 | 57.9 | – |
| Open3DIS | Offline | GroundedSAM | 29.9 | 46.7 | 58.6 | – |
| SAI3D | Offline | SemanticSAM | 30.8 | 50.5 | 70.6 | – |
| SAM3D | Online | SAM | 20.2 | 34.0 | 53.3 | 1369+1518 |
| ESAM | Online | SAM | 48.2 | 70.3 | 85.3 | 1369+**80** |
| ESAM-E | Online | FastSAM | **49.3** | **71.4** | **85.8** | 20+**80** |

Table 8: Decomposition of the inference time (ms) of ESAM excluding VFM.

| Backbone | | Decoder | Merging | | | Total |
|---|---|---|---|---|---|---|
| 3D-Unet | Adapters | | Similarity | Matching | Updating | |
| 41.0 | 28.0 | 5.0 | 0.7 | 0.3 | 5.0 | 80 |

## A.5 VISUALIZATION ON GEOMETRIC-AWARE POOLING

We provide the visualization of point-wise weights predicted in Geometric-aware Pooling in Figure 7. The points with high weights are shown in red, while ones with low weights are shown in blue. It is shown that noisy boundary with inaccurate shape information will be assigned low weight, while regions contain objects (especially small objects) are assigned high weight for better segmentation.

## A.6 QUALITATIVE RESULTS ON SCENENN AND 3RSCAN

We further visualize different methods on SceneNN and 3RScan in Figure 8. Consistent with Figure 4, ESAM predicts accurate and clean 3D masks. On the contrary, the prediction of SAM3D is noisy or incomplete while the prediction of SAI3D tends to be over-segmented.

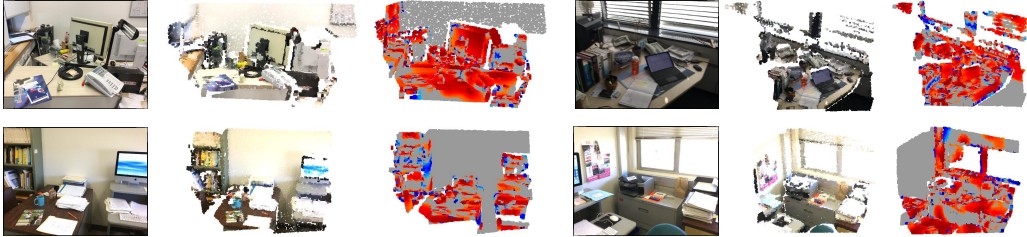

Figure 7: Visualization of weights in Geometric-aware Pooling. We also provide the corresponding RGB images. The points with high weights are shown in red, while ones with low weights are shown in blue.

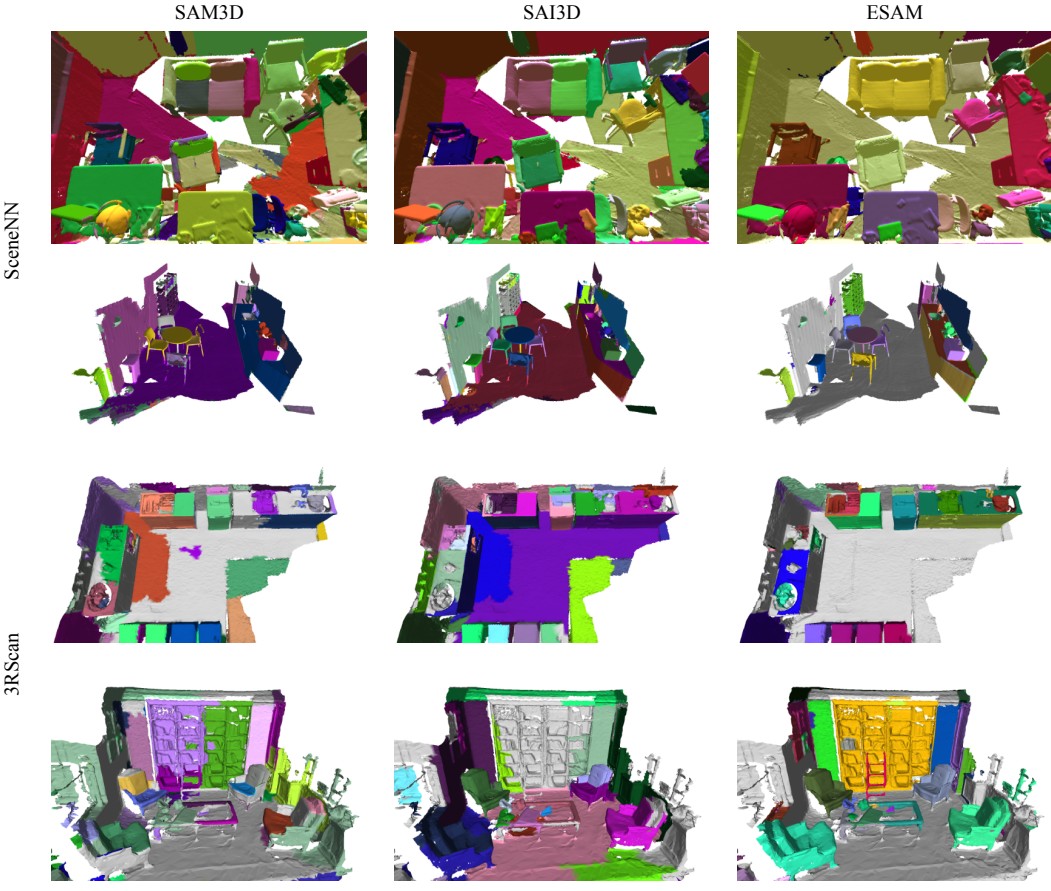

Figure 8: Visualization results on SceneNN and 3RScan.

