# OpenReview forum: "EmbodiedSAM: Online Segment Any 3D Thing in Real Time"
_ICLR.cc/2025/Conference — ICLR 2025 Oral_

### Official Review · Reviewer_jHbV · 2024-10-30

**Soundness:** 3
**Presentation:** 4
**Contribution:** 3
**Rating:** 8
**Confidence:** 5

**Summary:**

The paper presents EmbodiedSAM, a framework designed for online 3D instance segmentation, leveraging the Segment Anything Model (SAM). This method integrates SAM with a novel merging strategy that fuses geometric, contrastive, and semantic information to deliver high-efficiency segmentation of 3D objects in dynamic environments. By enabling real-time segmentation, EmbodiedSAM is positioned to enhance 3D perception tasks where immediate processing is critical. The framework is evaluated on several datasets, demonstrating its potential to perform effectively in real-world applications where timely response is essential.

**Strengths:**

1. This paper is well-written and easy to follow.
2. The experiment part of this paper is sufficient.
3. This paper provides a nice summary of existing works.

**Weaknesses:**

This paper is very interesting, I think it has reached the standard for acceptance at the conference, I only have a few simple questions.

1. The Geo-aware Pooling module is a common technology. So, I think it should not be one of the main contributions of this paper, the whole pipeline is the most important design.
2. As an online segmentation network, the author should report the time cost of each module, such as super point building, query refinement, and merging.
3. The results presented in Table 6 should include more detailed explanations regarding the performance impact of removing each of the three auxiliary tasks (geometric, contrastive, and semantic similarities) in the merging strategy. This clarification is necessary to understand the individual contribution of each auxiliary task.
4. The ablation study is conducted across different datasets, making it difficult to directly compare the results. It would be more effective to perform the entire ablation study on a single dataset to ensure clearer and more consistent analysis.

**Questions:**

See the "weakness" part.

---

> ### Author Response · Authors · 2024-11-21
>
> Thank you for your valuable comments and kind words to our work. Below we address specific questions.
>
> **Q1: About the novelty of Geo-aware Pooling**
>
> We agree with the reviewer that PointNet-like architecture of Geo-aware Pooling is not new. But we think this is a new technique for superpoint pooling. Previous method like SPFormer and Oneformer3D directly use average pooling condense point features to superpoints without considering the shape of each superpoint. We believe this operation is sub-optimal and propose an efficient module to predict the weights of each point for better pooling. We think weighted average is a better way to obtain superpoint features, which may be helpful for future works.
>
> **Q2: Time cost of each module**
>
> We have reported the time cost of each module in Appendix A.3. The 3D part of EmbodiedSAM takes 80ms per frame, while the 2D part (VFM for superpoint building) takes 20ms per frame.
>
> **Q3: More detailed explanations on results in Table 6**
>
> Thanks for the kind suggestion. We notice that removing the box representation results in the most significant performance degradation. This is because most cases of merging can be decided based on geometry. In fact, if two 3D masks are very close to each other, or the whole geometry of them are likely to be overlapped, they are likely to belong to the same object. We also observe that the influence of removing contrastive representation is greater than removing semantic representation. This is because contrastive similarity is designed for instance merging, while semantic similarity is originally designed for mask classification.
>
> **Q4: Ablation on a single dataset**
>
> Since query lifting and query decoder are not related to inter-frame merging, we conduct ablation on them on ScanNet-25k dataset in Table 5. The performance is evaluated on each single RGB-D image, rather than a whole RGB-D sequence. Following your advice, we further conduct ablation studies of query lifting and query decoder on ScanNet sequences, as shown below:
>
> | Method | AP | Latency (ms) |
> | :-------- | :---------: | :---------: |
> | Replace $\mathcal{G}$ with Average Pooling | 38.0 | 78 |
> | Set $F=F_S$ only | 30.3 | 76 |
> | Set $F=F_P$ only | 41.5 | 88 |
> | The final model | 41.6 | 80 |
>
> It is shown that these two methods can significantly improve the performance of EmbodiedSAM with negligible computational overhead.
>
> **If you have any further question, please let us know. Thank you very much!**

---

> > ### Comment · Reviewer_jHbV · 2024-11-25
> > **Official Comment by Reviewer jHbV**
> >
> > I do not have any other questions. Thank the authors for the excellent work; I will keep my score.
> > Do not reply, thanks!

---

### Official Review · Reviewer_H4Wa · 2024-11-04

**Soundness:** 3
**Presentation:** 3
**Contribution:** 3
**Rating:** 8
**Confidence:** 4

**Summary:**

The paper proposes a 3D instance segmentation method that is online, real-time, fine-trained and generalizable. Specifically it lifts the 2D segmentation masks generated by SAM to 3D using depth and uses the instance indices to group points into superpoints. It proposes a dula-level query decoder to use both points and superpoints for efficiently generating fine-grained masks. As an online method, it proposes a efficient way to merge masks at current timestamp to previous masks, which considers geometric similarity, contrastive similarity and semantic similarity. Because each superpoint has a feature vector, the similarities can be compuated as matrix operation for quick inference. Experiments were conducted on four baseline datasets and the proposed method outperforms both previous online methods and offline methods, and it also shows generalizability on zero-shot cases.

**Strengths:**

1. The paper proposes a novel way to lift 2D segmentation masks to 3D that is geometry aware. It also proposes an algorithm to better, efficiently merge 3D masks from different timestamps.
2. The method can achieve better results on four datasets, most notably in a real-time way (can achieve 10 fps with FastSAM as backbone). This is very important for its application in real-world scenarios.
3. The paper is well-organized and easy to follow.

**Weaknesses:**

1. All qualitative results are on ScanNet200. Wondering if the proposed method also has clear visual improvements on other datasets used in the paper.
2. Since the method is targeted to be used in real-world embodied tasks and is supposed to generalize to unseen scenes, it would be more convincing if results on some real-world captures are included.
3. For generating superpoint features in equation (3), why the $z^{global}$ is added instead of concatenated to the pooled point features? Also why using the original point features instead of $z^{local}$

**Questions:**

1. Typo: L280: position pair-> positive pair
2. All results included in the paper are class-agnostic. I am curious if the proposed method has advantages over all object categories equally or it works better on certain categories.

---

> ### Author Response · Authors · 2024-11-21
>
> Thank you for your valuable comments and kind words to our work. Below we address specific questions.
>
> **Q1: Visualization on other datasets**
>
> We further visualize different methods on SceneNN and 3RScan, as shown in Appendix A.6. Consistent with Figure 4, ESAM predicts accurate and clean 3D masks. On the contrary, the prediction of SAM3D is noisy or incomplete while the prediction of SAI3D tends to be over-segmented.
>
> **Q2: Experiments on real-world captures**
>
> EmbodiedSAM requires posed RGB-D stream as input, where RGB-D images can be captured by a RGB-D camera and camera pose can be online estimated by SLAM or visual odometry system. Therefore, to conduct real-world experiments, we should first build an online 3D reconstruction system and then apply EmbodiedSAM simultaneously with camera tracking. Since it will take us some time to buy accurate RGB-D camera, set up the whole hardware and software system, calibrate the camera and select proper hyperparamters to filter noisy depth, we may not be able to finish these tasks before the rebuttal due.
>
> Nevertheless, since the RGB-D sequences used in our paper are all captured with real-world sensors, we can evaluate the real-world performance of EmbodiedSAM by adding noise to the process of camera tracking. We add different scales of noise (random noise, with a maximum ratio) on the camera extrinsic parameters and run EmbodiedSAM on ScanNet200-class-agnostic following the setting of Table 1. The results are shown below:
>
> | Noise Ratio | AP | AP50 | AP25 |
> | :--------: | :---------: | :---------: | :---------: |
> | 0% | 43.4 | 65.4 | 80.9 |
> | 1% | 43.5 | 66.2 | 80.8 |
> | 5% | 43.3 | 66.1 | 80.6 |
> | 10% | 41.3 | 63.5 | 79.3 |
>
> It is shown that our method is robust to the noise on camera pose, which is a huge advantage when applied in real-world applications.
>
> **Q3: Why $z^{global}$ is added instead of concatenated to the pooled point features**
>
> We empirically find adding the features can achieve slightly better performance than concatenating them. Moreover, adding is also more computationally efficient than concatenating since it does not require a linear layer for channel projection.
>
> **Q4: Why using the original point features instead of $z^{local}$**
>
> The nature of geometric-aware pooling is a weighted average of point features inside each superpoint. $z^{local}$ is only used to predict the weights of each point for superpoint pooling. It is not a better geometric representation than the original point features. We empirically find that the overall performance decreases by about 0.7% if we use $z^{local}$ instead of the original point features.
>
> **Q5: Any class-aware experiments**
>
> We need to clarify that the experiments in Table 3 are class-aware. There are 18 classes in ScanNet and 15 classes in SceneNN. We report the per-category performance of TD3D-MA and our method on ScanNet as below:
>
> AP25:
>
> |  | cabinet | bed | chair | sofa | table | door | window | bookshelf | picture | counter | desk | curtain | fridge | curtain | toilet | sink | bathtub | others | mean |
> | :--------: | :---------: | :---------: | :---------: |  :---------: | :---------: | :---------: | :---------: | :---------: | :---------: | :---------: | :---------: | :---------: | :---------: | :---------: | :---------: | :---------: | :---------: | :---------: | :---------: |
> | TD3D-MA | 60.3 | 86.8 | 91.5 | 80.3 | 72.8 | 56.0 | 55.3 | 67.5 | 45.1 | 48.9 | 72.9 | 68.4 | 56.5 | 86.3 | 99.7 | 81.3 | 87.8 | 65.3 | 71.3 |
> | ESAM-E+FF | 69.8 | 79.3 | 91.9 | 81.3 | 80.4 | 78.7 | 68.1 | 73.5 | 67.9 | 59.4 | 72.1 | 73.6 | 61.2 | 75.1 | 100.0 | 88.2 | 94.6 | 73.1 | 77.1 |
>
> AP50:
>
> |  | cabinet | bed | chair | sofa | table | door | window | bookshelf | picture | counter | desk | curtain | fridge | curtain | toilet | sink | bathtub | others | mean |
> | :--------: | :---------: | :---------: | :---------: |  :---------: | :---------: | :---------: | :---------: | :---------: | :---------: | :---------: | :---------: | :---------: | :---------: | :---------: | :---------: | :---------: | :---------: | :---------: | :---------: |
> | TD3D-MA | 50.9 | 79.1 | 82.5 | 71.3 | 63.6 | 44.0 | 36.0 | 45.5 | 38.5 | 30.3 | 57.3 | 49.8 | 52.9 | 78.9 | 99.7 | 66.6 | 84.9 | 56.9 | 60.5 |
> | ESAM-E+FF | 50.5 | 70.5 | 84.0 | 64.4 | 73.5 | 62.8 | 48.1 | 42.8 | 60.2 | 34.7 | 43.8 | 53.1 | 51.4 | 65.4 | 100.0 | 65.3 | 81.7 | 62.4 | 61.9 |
>
> **If you have any further question, please let us know. Thank you very much!**

---

> > ### Comment · Reviewer_H4Wa · 2024-11-26
> >
> > Thanks to the authors for the answers. I don't have further questions. I'll raise my score.

---

### Official Review · Reviewer_tbzg · 2024-11-07

**Soundness:** 3
**Presentation:** 3
**Contribution:** 3
**Rating:** 8
**Confidence:** 2

**Summary:**

This paper proposes a method for online 3D instance segmentation. Given a sequence of posed RGB-D images, they first use SAM to acquire 2D masks, and unproject them to 3D superpoints using the depth maps. Then they propose a novel framework to aggregate geometric features of superpoints, followed by a dual-level query decoder to predict 3D masks. They additionally propose to leverage bounding boxes IOU and semantics to assist the similarity computation, and use feature contrasting to further improve the performance.

**Strengths:**

1. Motivation: The paper proposes a novel framework for online, efficient 3D instance segmentation from RGB-D sequences, which is of high potential for embodied tasks.
2. Contribution: The proposed framework is efficient in mask generation and query merging, and the use of the IOU and semantic are reasonable and effective.
3. Performance: The performance greatly surpasses the baselines.

**Weaknesses:**

I'm not very familiar with the task of this paper and did not find obvious weaknesses in this paper. Please see the questions below.

**Questions:**

1. According to Table 5 and 6, the performance gains mainly come from the merging strategies. An ablation of the geometric-aware query lifting and dual-level query decoder would better evaluate the necessity of the proposed framework.
2. According to Table 6, the computation of the similarity matrix in Eq.(7) plays a vital role, and it would be beneficial if can provide an ablation study on the weights of the different terms in Eq.(7). Besides, is directly adding them better than multiplying them?

---

> ### Author Response · Authors · 2024-11-21
>
> Thank you for your valuable comments and kind words to our work. Below we address specific questions.
>
> **Q1: An ablation of the geometric-aware query lifting and dual-level query decoder**
>
> Since query lifting and query decoder are not related to inter-frame merging, we conduct ablation on them on ScanNet-25k dataset in Table 5. The performance is evaluated on each single RGB-D image, rather than a whole RGB-D sequence. Following your advice, we further conduct ablation studies of query lifting and query decoder on ScanNet sequences, as shown below:
>
> | Method | AP | Latency (ms) |
> | :-------- | :---------: | :---------: |
> | Replace $\mathcal{G}$ with Average Pooling | 38.0 | 78 |
> | Set $F=F_S$ only | 30.3 | 76 |
> | Set $F=F_P$ only | 41.5 | 88 |
> | The final model | 41.6 | 80 |
>
> It is shown that these two methods can significantly improve the performance of EmbodiedSAM with negligible computational overhead.
>
> **Q2: Ablation study on the weights of the different terms in Eq.(7). Adding or multiplying.**
>
> An ablation study on the weights for geometric, contrastive and semantic similarity is shown below (following the setting of Table 3):
>
> | Weights | AP | AP50 | AP25 |
> | :--------: | :---------: | :---------: | :---------: |
> | 0.33/0.33/0.33 | 39.3 | 57.5 | 73.1 |
> | 0.5/0.25/0.25 | 38.5 | 56.3 | 72.4 |
> | 0.25/0.5/0.25 | 39.6 | 58.0 | 73.2 |
> | 0.25/0.25/0.5 | **41.6** | **60.1** | **75.6** |
>
> We empirically find adding these terms is more robust than multiplying them. The latter is more sensitive to the hyperparameters and may get worse performance when we apply EmbodiedSAM to new environments.
>
> **If you have any further question, please let us know. Thank you very much!**

---

> ### Author Response · Authors · 2024-11-26
>
> Dear reviewer:
>
> Since the discussion stage is nearing its end, we would appreciate your feedback and are happy to address any concerns you may have.

---

### Official Review · Reviewer_bGw4 · 2024-11-08

**Soundness:** 3
**Presentation:** 3
**Contribution:** 3
**Rating:** 6
**Confidence:** 4

**Summary:**

This paper proposes a geometry-aware query lifting module that represents 2D masks generated by SAM as 3D-aware queries. These queries are then iteratively refined using a similarity matrix between 3D masks from various viewpoints through a dual-level query decoder. EMBODIEDSAM partially addresses the offline and slow-processing issues present in most existing VFM-assisted 3D perception methods, making it more suitable for real-world embodied tasks.

**Strengths:**

1. Using InsQuery instead of traditional handcrafted strategies achieves faster and geometrically consistent multi-frame mask merging, ultimately enabling real-time inference speed.
2. The paper is well-structured and clearly written.
3. The supplementary video demo effectively highlights the online, real-time, fine-grained, and highly generalized capabilities of EmbodiedSAM.

**Weaknesses:**

1. How is recognition achieved for any 3D object category? Why are the results in Table 1 for class-agnostic 3D instance segmentation？
2. In Figure 4, it would be helpful to match the colors of predicted and ground truth instances for easier comparison.
3. Provide additional comparisons with more recent offline methods, such as OpenMask3D[1] and OpenIns3D[2].

Reference:

[1] Takmaz, Ayça, et al. "Openmask3d: Open-vocabulary 3d instance segmentation." arXiv preprint arXiv:2306.13631 (2023).

[2] Huang, Zhening, et al. "Openins3d: Snap and lookup for 3d open-vocabulary instance segmentation." arXiv preprint arXiv:2309.00616 (2023).

**Questions:**

See the weakness part.

---

> ### Author Response · Authors · 2024-11-21
>
> Thank you for your valuable comments and kind words to our work. Below we address specific questions.
>
> **Q1: How to obtain category? Why Table 1 for class-agnostic 3D instance segmentation**
>
> We need to clarify that our method is a segment anything model, whose output is class-agnostic 3D masks. We follow the same setting with other segment anything models, like SAM3D and SAI3D. But our method is also able to predict category for each mask, which can be easily done by adding a MLP upon the 3D queries of each mask to predict the corresponding category. The class-aware instance segmentation results (reported in Table 3) are just predicted in this way.
>
> **Q2: Revision on Figure 4**
>
> We appreciate your constructive suggestion. However, since the prediction of 3D instance is not well-ordered, the color for each mask cannot be easily set to the same as the ground-truth. Other segment anything methods like SAM3D and SAI3D also suffer from this problem. We will try to design a more suitable visualization paradigm for 3D instance segmentation for easier comparison between prediction and ground-truth. Due to time constraints, we aim to revise our figure in the final version of this paper.
>
> **Q3: Compare with OpenMask3D and OpenIns3D**
>
> OpenMask3D and OpenIns3D are two representative open-vocabulary 3D instance segmentation methods, which focus on how to assign open-vocabulary labels on each 3D mask. These two methods adopt off-the-shelf 3D segmentation model and study how to classify each mask. Therefore, these two methods are classification approach, rather than segmentation approach.
>
> However, it is very valuable to combine these two kinds of approaches. We have conducted experiments to adopt EmbodiedSAM for 3D mask generation and adopt OpenMask3D for mask classification. In this way, we can achieve open-vocabulary and online 3D instance segmentation. We compare the open-vocabulary results with SAI3D as below:
>
> | Method | AP | AP50 | AP25 |
> | :-------- | :---------: | :---------: | :---------: |
> | SAI3D | 9.6 | 14.7 | 19.0 |
> | ESAM | **13.7** | **19.2** | **23.9** |
>
> We have also cited and discussed with these two papers in Appendix A.4.
>
> **If you have any further question, please let us know. Thank you very much!**

---

> ### Author Response · Authors · 2024-11-26
>
> Dear reviewer:
>
> Since the discussion stage is nearing its end, we would appreciate your feedback and are happy to address any concerns you may have.

---

> ### Comment · Reviewer_bGw4 · 2024-11-26
>
> Thank you for your response. I trust that the authors will revise Figure 4, and I am willing to accept this work. Additionally, I appreciate your efforts in producing such excellent work.

---

> > ### Author Response · Authors · 2024-11-26
> >
> > Thanks for your positive feedback! We will continue to polish our work and release the code. The instance color in FIgure 4 will be refined in the final version.

---

### Official Review · Reviewer_qwSP · 2024-11-08

**Soundness:** 3
**Presentation:** 3
**Contribution:** 2
**Rating:** 8
**Confidence:** 4

**Summary:**

This paper introduces a real-time, online 3D instance segmentation method for embodied tasks, leveraging the Segment Anything Model (SAM) to address the lack of high-quality 3D data for direct training. Unlike existing methods, which are often too slow for practical use, this approach translates SAM’s 2D masks into fine-grained 3D shapes via a geometric-aware query lifting module and a dual-level query decoder. This enables efficient object matching across frames, achieving state-of-the-art results on benchmarks like ScanNet and demonstrating strong generalization in zero-shot and data-efficient scenarios.

**Strengths:**

1. The writing is clear and easy to follow.
2. The method is technically sound.
3. The performance results are strong.

**Weaknesses:**

1. The camera's intrinsic and extrinsic parameters are crucial for this task. In embodied environments, it is very important to obtain accurate and stable intrinsic and extrinsic parameters. How can you acquire them in embodied environments? Whether the unstable and incorrect  intrinsic and extrinsic parameters affect the final results? Additionally, in lines 139-140, the point clouds are generated by projecting the depth image into 3D space using pose parameters. So, how are two point clouds aligned and merged? In real-world scenarios, point cloud alignment is also a significant issue.

2. There is a lack of visualization for the point-wise weights predicted in Geometric-aware Pooling. Providing this visualization would clarify how the model weights features across different geometric regions.

3. Regarding the class-agnostic 3D instance segmentation results compared in Table 1, the fairness of the comparison could be questioned. Since Open3DIS does not use semantic and instance labels for supervision, it may not be entirely fair to compare it with fully-supervised methods. I think a comparison with fully-supervised online 3D instance segmentation methods, as done in Table 3, would be more appropriate.

**Questions:**

See weakness above.

---

> ### Author Response · Authors · 2024-11-21
>
> Thank you for your valuable comments and kind words to our work. Below we address specific questions.
>
> **Q1: How to acquire accurate and stable intrinsic and extrinsic parameters in embodied environments**
>
> The input to EmbodiedSAM is streaming RGB-D video, so an online camera pose tracking method should be applied to obtain extrinsic parameters. Many SLAM methods can solve this problem, such as DROID-SLAM [1], DPVO [2] and SplaTAM [3]. The intrinsic parameters can be obtained by camera calibration. After calibration, we treat them as a constant matrix, as done in our experiments.
>
> **Q2: Robustness to noise**
>
> We add different scales of noise (random noise, with a maximum ratio) on the camera extrinsic parameters and run EmbodiedSAM on ScanNet200-class-agnostic following the setting of Table 1. The results are shown below:
>
> | Noise Ratio | AP | AP50 | AP25 |
> | :--------: | :---------: | :---------: | :---------: |
> | 0% | 43.4 | 65.4 | 80.9 |
> | 1% | 43.5 | 66.2 | 80.8 |
> | 5% | 43.3 | 66.1 | 80.6 |
> | 10% | 41.3 | 63.5 | 79.3 |
>
> It is shown that our method is robust to the noise on camera pose. As for intrinsic parameters, since the camera calibration technology is very mature, the error is negligible. All our experiments on the same dataset use a constant matrix obtained by camera calibration as the camera intrinsic.
>
> **Q3: How to align point clouds**
>
> Since the camera extrinsic parameters can be estimated as described in Q1, we do not need to align the point clouds of different frames. We can use camera intrinsic and extrinsic parameters to convert depth image into point clouds in world coordinate system and directly concatenate them.
>
> **Q4: Visualization of weights in Geometric-aware Pooling**
>
> We have provided the visualization of point-wise weights predicted in Geometric-aware Pooling in Appendix A.5. It is shown that noisy boundary with inaccurate shape information will be assigned low weight, while regions contain objects (especially small objects) are assigned high weight for better segmentation.
>
> **Q5: Comparison with Zero-shot Methods**
>
> We appreciate your constructive suggestion. We agree that EmbodiedSAM uses semantic and instance labels for supervision while some baselines do not. However, the dataset transferring experiments in Table 2 validate the generalization ability of EmbodiedSAM. Following your advice, we further implement TD3D-MA and compare with it following the setting of Table 1, as shown below:
>
> | Method | AP | AP50 | AP25 |
> | :-------- | :---------: | :---------: | :---------: |
> | TD3D-MA | 22.9 | 30.4 | 35.1 |
> | ESAM-E | **43.4** | **65.4** | **80.9** |
>
> It is shown that EmbodiedSAM significantly outperforms fully-supervised online 3D instance segmentation method. This is because our VFM-assisted prediction pipeline can better segment small objects, which is a bottleneck for other fully-supervised online 3D instance segmentation methods.
>
> [1] DROID-SLAM: Deep Visual SLAM for Monocular, Stereo, and RGB-D Cameras. NeurIPS 2021.
>
> [2] Deep Patch Visual Odometry. NeurIPS 2023.
>
> [3] SplaTAM: Splat, Track & Map 3D Gaussians for Dense RGB-D SLAM. CVPR 2024.
>
> **If you have any further question, please let us know. Thank you very much!**

---

> > ### Comment · Reviewer_qwSP · 2024-11-22
> >
> > Thank you for your response. Some of my concerns have been addressed, but there are still a few questions that require further clarification:
> >
> > **Q2: Robustness to Noise**
> >
> > In this section, the authors introduced noise (random noise with a maximum ratio) to the camera extrinsic parameters. Could you clarify how the metrics are evaluated in this context? It seems that the correspondence between the merged point clouds and the ground truth point clouds becomes ambiguous under noise. As mentioned in Line 304, the losses are computed on a per-frame basis. Does this mean the metrics are evaluated at the frame level?
> >
> > **Q3: Alignment of Point Clouds**
> >
> > When aligning point clouds, directly concatenating them could alter the density in overlapping regions (different from non-overlapping regions). Would this affect the segmentation results?
> >
> > **Q4: Visualization of Weights in Geometric-aware Pooling**
> >
> > It would be helpful to provide visualized, color-coded point clouds, as this would allow for a clearer observation and better understanding of the weight distribution.
> >
> > **Q5: Comparison with Zero-shot Methods**
> >
> > A detailed per-class comparison on ScanNet200 would provide stronger evidence to support your claim that the proposed method performs better in segmenting small objects. I understand that listing all 200 categories might be impractical. Perhaps you could selectively present a subset of representative categories, particularly those with small objects, to highlight the advantages of your approach.

---

> > > ### Author Response · Authors · 2024-11-23
> > >
> > > Thanks for your prompt reply! We answer the questions as below:
> > >
> > > **Q2: Robustness to Noise**
> > >
> > > After adding noise, the metrics are evaluated the same as done in Table 1, which is detailed in Line 372-375. We use nearest neightbor interpolation to map the predicted instance labels to the ground-truth point clouds for evaluation.
> > >
> > > EmbodiedSAM only requires frame-level supervision, but its performance is evaluated at scene level. All experiments on ScanNet, ScanNet200, SceneNN and 3RScan compute the metrics on complete 3D scenes.
> > >
> > > **Q3: Alignment of Point Clouds**
> > >
> > > Yes, directly concatenating point clouds from different frames will alter the density in overlapping regions. However, since EmbodiedSAM adopts sparse convolutional backbone to extract features, it is robust to different point cloud densities.
> > >
> > > The change of density can be observed in our video demo or Figure 6. It is shown that EmbodiedSAM works well on overlapped regions.
> > >
> > > **Q4: Visualization of Weights in Geometric-aware Pooling**
> > >
> > > Thanks very much for your kind suggestion. We further provide the visualization of point clouds in original color in Figure 7 for clear demonstration.
> > >
> > > **Q5: Comparison with Zero-shot Methods**
> > >
> > > We select some representative categories and report performance on them as below, where categories after refrigerator (pillow, picture, box, ...) are small objects:
> > >
> > > AP25：
> > >
> > > |         | chair | table | door | cabinet | shelf | desk | bed  | toilet | window | refrigerator | **pillow** | **picture** | **box** | **towel** | **bag** | **microwave** | **stove** | **mirror** | **copier** | **fan** | **suitcase** | **clock** | laundry **basket** | **jacket** | **toilet paper dispenser** | **bal**l | **hat** | **decoration** | **mailbox** | **ceiling light** | **trash bin** | **poster** |
> > > | :-----: | :---: | :---: | :--: | :-----: | :---: | :--: | :--: | :----: | :----: | :----------: | :--------: | :---------: | :-----: | :-------: | :-----: | :-----------: | :-------: | :--------: | :--------: | :-----: | ------------ | :-------: | :----------------: | :--------: | :------------------------: | :------: | :-----: | :------------: | :---------: | :---------------: | :-----------: | :--------: |
> > > | TD3D-MA | 67.2  | 56.6  | 57.8 |  50.6   | 34.5  | 55.3 | 56.3 |  63.7  |  49.8  |     53.1     |    3.3     |    50.8     |  12.5   |    9.9    |   6.7   |      4.8      |   14.5    |    17.2    |    23.8    |  21.0   | 12.8         |    9.1    |        25.0        |    5.6     |            13.9            |   50.0   |  33.3   |      7.4       |    55.9     |        9.3        |     63.0      |    42.9    |
> > > | ESAM-E  | 93.4  | 85.0  | 87.4 |  81.3   | 68.2  | 85.6 | 83.5 |  98.3  |  70.4  |     94.5     |    82.6    |    74.7     |  73.7   |   76.7    |  88.9   |     95.2      |   84.2    |    63.8    |    95.4    |  79.3   | 100.0        |   81.8    |       100.0        |    82.2    |            91.7            |  100.0   |  100.0  |      75.6      |    88.0     |       68.7        |     98.9      |    57.1    |
> > >
> > > AP50：
> > >
> > > |         | chair | table | door | cabinet | shelf | desk | bed  | toilet | window | refrigerator | **pillow** | **picture** | **box** | **towel** | **bag** | **microwave** | **stove** | **mirror** | **copier** | **fan** | **suitcase** | **clock** | **laundry basket** | **jacket** | **toilet paper dispenser** | **ball** | **hat** | **decoration** | **mailbox** | **ceiling light** | **trash bin** | **poster** |
> > > | :-----: | :---: | :---: | :--: | :-----: | :---: | :--: | :--: | :----: | :----: | :----------: | :--------: | :---------: | :-----: | :-------: | :-----: | :-----------: | :-------: | :--------: | :--------: | :-----: | :----------: | :-------: | :----------------: | :--------: | :------------------------: | :------: | :-----: | :------------: | :---------: | :---------------: | :-----------: | :--------: |
> > > | TD3D-MA | 64.9  | 50.6  | 55.0 |  44.7   | 25.5  | 48.2 | 47.7 |  63.7  |  41.9  |     53.1     |    2.1     |    47.5     |   8.0   |    6.8    |   6.7   |      4.8      |    0.0    |    10.7    |    3.4     |   8.6   |     12.8     |    9.1    |        9.4         |    2.4     |            5.6             |   50.0   |  33.3   |      7.4       |    46.5     |        4.0        |     57.4      |    9.5     |
> > > | ESAM-E  | 82.3  | 77.8  | 78.3 |  68.7   | 46.9  | 57.1 | 76.8 |  95.6  |  43.1  |     88.7     |    77.8    |    67.3     |  52.4   |   67.1    |  85.1   |     85.7      |   72.3    |    27.8    |    95.4    |  84.9   |     96.0     |   76.7    |       100.0        |    66.6    |            49.3            |  100.0   |  100.0  |      60.5      |    83.8     |       56.3        |     93.3      |    57.1    |
> > >
> > >
> > > **Hope our reply can solve your problem. If you have any further question, please let us know.**

---

> > > > ### Comment · Reviewer_qwSP · 2024-11-24
> > > >
> > > > Thank you for your quick response. I do not have any other concerns. I think this paper is above the acceptance threshold. If other reviewers do not have significant concerns, I will raise my score.

---

> > > > > ### Author Response · Authors · 2024-11-24
> > > > >
> > > > > We appreciate the reviewer for the positive feedback. Your constructive comments and suggestions are indeed helpful for improving the paper. We will continue to polish our work and release the code.

---

> > > > > ### Author Response · Authors · 2024-11-28
> > > > >
> > > > > Dear Reviewer qwSP,
> > > > >
> > > > > As the rebuttal phase draws to a close, we would like to express our gratitude for your thoughtful feedback on our paper. We have addressed all reviewers’ comments, including yours, and no additional concerns have been raised.
> > > > >
> > > > > At this stage, Reviewer eVni and Reviewer H4Wa have updated their scores to 8, while Reviewer tbzg and Reviewer jHbV initially gave a score of 8 and maintained it. Additionally, both Reviewer bGw4 and Reviewer jHbV have evaluated our work as 'excellent'. Based on this, we believe that all significant concerns have been fully resolved.
> > > > >
> > > > > Therefore, we would kindly like to ask if you might consider adjusting your score to better reflect your updated assessment of the paper. Your support would mean a lot to us and contribute significantly to the final evaluation process.
> > > > >
> > > > > Thank you once again for your thoughtful review and contributions to improving our work.

---

> > > > > > ### Comment · Reviewer_qwSP · 2024-12-02
> > > > > >
> > > > > > Thank you for your strong rebuttal. I do not have any other questions and have raised my score to 8.

---

### Official Review · Reviewer_eVni · 2024-11-09

**Soundness:** 3
**Presentation:** 2
**Contribution:** 3
**Rating:** 8
**Confidence:** 2

**Summary:**

This paper leverages SAM for 3D instance segmentation, incorporating a geometry-aware query lifting module that transforms 2D masks into fine-grained 3D shapes on point clouds. Using 3D queries, it efficiently merges 3D masks across frames through simple matrix operations. This design enables online, real-time 3D instance segmentation.

**Strengths:**

Unlike approaches that apply SAM on individual images and project 2D masks onto 3D point clouds, this paper introduces a geometry-aware module that lifts 2D masks to 3D queries. This enables the prediction of temporally and geometrically consistent 3D masks using iterative query refinement.

This work eliminates the need for handcrafted merging strategies by instead identifying similarities between newly predicted and previously generated 3D masks, resulting in improved performance.

Achieving real-time inference, and outperforming not only online methods but also offline methods.

**Weaknesses:**

The abstract and introduction quickly dive into the implementation without providing sufficient motivation to explain why this approach is taken. For instance, it lacks clarity on how 3D-aware is achieved, the role of queries, the need for iterative refinement, and the insight of the dual-level query decoder. Although the comparison with prior work is clear, the explanation of the proposed method lacks an overview, leaving out insights behind the approach.

**Questions:**

Lack of discussion with respect to :
Gaussian Grouping: Segment and Edit Anything in 3D Scenes
LangSplat: 3D Language Gaussian Splatting
Segment Any 3D Gaussians

---

> ### Author Response · Authors · 2024-11-21
>
> Thank you for your valuable comments and kind words to our work. Below we address specific questions.
>
> **Q1: An overview of the proposed method**
>
> We thanks for your suggestion on the introduction of our approach. The main idea of our architecture is to lift the 2D segmentation results generated by SAM to 3D queries for fast and robust merging. Previous methods like SAM3D directly project 2D mask to 3D point clouds, followed by mask-wise merging operation to unify masks of the same instance across frames. This merging operation is very slow and not robust due to the handcrafted criterion. On the contrary, our EmbodiedSAM adopts a learnable projection to convert 2D mask to 3D queries, which converts the mask merging problem into query merging. In this way, we can utilize matrix operation to compute the similarity between masks and merge them in parallel.
>
> Specifically, 3D-aware is achieved by our 3D backbone and geometric-aware pooling. The role of query is to represent 3D mask in a fixed-length vector for efficient mask merging. The iterative refinement is designed to extract better representation for each 3D mask. The dual-level decoder is designed to balance the computational cost and granularity of mask generation, as described in Line 223-Line 233.
>
> **Q2: Discussion with Gaussian Grouping and LangSplat**
>
> Gaussian Grouping and LangSplat are two representative offline segmentation methods for 3D gaussian. They focus on how to accurately segment the 3D scene represented by 3D gaussian. This is a different setting from EmbodiedSAM, since we focus on how to online segment anything for streaming RGB-D video. We have cited these two papers and discussed with them in our related work.
>
> **If you have any further question, please let us know. Thank you very much!**

---

> ### Author Response · Authors · 2024-11-26
>
> Dear reviewer:
>
> Since the discussion stage is nearing its end, we would appreciate your feedback and are happy to address any concerns you may have.

---

> > ### Comment · Reviewer_eVni · 2024-11-26
> >
> > No further concerns. Based on your response and other reviews, I’ll raise my score.

---

### Author Response · Authors · 2024-11-21

Thanks for all reviewers’ constructive comments and patience on our work. We are glad to receive positive feedbacks from all reviewers. We have provided detailed replies to the questions of each reviewer and refined our paper following the suggestions. The modification on our paper mainly including:
- Open-vocabulary 3D instance segmentation results in Appendix A.4.
- Visualization of point-wise weights predicted in Geometric-aware Pooling in Appendix A.5.
- Qualitative results on other datasets except for ScanNet200 in Appendix A.6.
- Some writing and presentation are improved.

---

### Meta-Review · Area_Chair_yhb7 · 2024-12-17

**Metareview:**

This paper receives unanimous positive ratings of 8,8,6,8,8. The AC follows the recommendations of the reviewers to accept the paper. The reviewers think that the proposed method of using a geometry-aware module that lifts 2D mask to 3D queries is effective, the performance of the proposed method is strong and the proposed method can work in real-time inference. The proposed method can even outperform offline methods. The weaknesses mentioned by the reviewers are mostly clarifications, which are well-addressed by the authors in the discussion and rebuttal phases.

**Additional Comments On Reviewer Discussion:**

The weaknesses mentioned by the reviewers are mostly clarifications, which are well-addressed by the authors in the discussion and rebuttal phases.

---

### Decision · Program_Chairs · 2025-01-22

Accept (Oral)